# Antiviral Activity of N_1_,N_3_-Disubstituted Uracil Derivatives against SARS-CoV-2 Variants of Concern

**DOI:** 10.3390/ijms231710171

**Published:** 2022-09-05

**Authors:** Andrei E. Siniavin, Mikhail S. Novikov, Vladimir A. Gushchin, Alexander A. Terechov, Igor A. Ivanov, Maria P. Paramonova, Elena S. Gureeva, Leonid I. Russu, Nadezhda A. Kuznetsova, Elena V. Shidlovskaya, Sergei I. Luyksaar, Daria V. Vasina, Sergei A. Zolotov, Nailya A. Zigangirova, Denis Y. Logunov, Alexander L. Gintsburg

**Affiliations:** 1N.F. Gamaleya National Research Center for Epidemiology and Microbiology, Ivanovsky Institute of Virology, Ministry of Health of the Russian Federation, 123098 Moscow, Russia; 2Department of Molecular Neuroimmune Signalling, Shemyakin-Ovchinnikov Institute of Bioorganic Chemistry, Russian Academy of Sciences, 117997 Moscow, Russia; 3Department of Pharmaceutical & Toxicological Chemistry, Volgograd State Medical University, 400131 Volgograd, Russia; 4Department of Virology, Lomonosov Moscow State University, 119991 Moscow, Russia; 5Department of Infectiology and Virology, Federal State Autonomous Educational Institution of Higher Education I M Sechenov First Moscow State Medical University of the Ministry of Health of the Russian Federation (Sechenov University), 119435 Moscow, Russia

**Keywords:** SARS-CoV-2, COVID-19, antiviral agents, non-nucleoside inhibitor, RNA-dependent RNA polymerase

## Abstract

Despite the widespread use of the COVID-19 vaccines, the search for effective antiviral drugs for the treatment of patients infected with SARS-CoV-2 is still relevant. Genetic variability leads to the continued circulation of new variants of concern (VOC). There is a significant decrease in the effectiveness of antibody-based therapy, which raises concerns about the development of new antiviral drugs with a high spectrum of activity against VOCs. We synthesized new analogs of uracil derivatives where uracil was substituted at the N_1_ and N_3_ positions. Antiviral activity was studied in Vero E6 cells against VOC, including currently widely circulating SARS-CoV-2 Omicron. All synthesized compounds of the panel showed a wide antiviral effect. In addition, we determined that these compounds inhibit the activity of recombinant SARS-CoV-2 RdRp. Our study suggests that these non-nucleoside uracil-based analogs may be of future use as a treatment for patients infected with circulating SARS-CoV-2 variants.

## 1. Introduction

The ongoing coronavirus disease 2019 (COVID-19) pandemic caused by severe acute respiratory syndrome coronavirus 2 (SARS-CoV-2) is a global public health crisis [1]. SARS-CoV-2 is a positive-sense, single-stranded RNA virus [2]. Like other coronaviruses [3], SARS-CoV-2 synthesizes a variety of viral enzymes and proteins that are essential for viral entry, replication, and pathogenesis, including structural and non-structural proteins [4]. As the pandemic progresses, several new variants of the virus emerged [5]. Five «variants of concern» (VOCs) (alpha, beta, gamma, delta, and omicron) have been described that are associated with increased virus transmission, virulence, or immune evasion (https://www.ecdc.europa.eu/en/COVID-19/variants-concern accessed on 28 April 2022). The development of effective antiviral drugs is a promising strategy [6]. Recently conducted randomized clinical trials identified antiviral drugs that target SARS-CoV-2. Remdesivir, molnupiravir, and nirmatrelvir have recently exhibited clinical benefits when administered early on during COVID-19 infection [7,8,9]. However, the number of substances with a proven antiviral effect against coronaviruses is still insufficient. Thus, as new variants of the virus with various mutations in the genome of the virus appear, resistance to antiviral drugs may develop. It is necessary to develop adequate combination therapy schemes and a reserve of active substances to counteract the loss of effectiveness in a timely manner.

One of the main molecular targets of viruses is their polymerase and replicative complex. Combinations of nucleoside and non-nucleoside reverse transcriptase inhibitors have been successfully used to treat HIV infection [10]. The main approach to the development of new antiviral drugs against SARS-CoV-2, like other viruses, is based on targeting RNA-dependent RNA polymerase (RdRp) [11,12]. For example, remdesivir is a potent nucleoside polymerase inhibitor of SARS-CoV-2 and MERS-CoV [13,14]. However, the mechanism of action of nucleoside inhibitors is based on chain termination and the mutation of viral progeny, which can lead to drug resistance and host genetic toxicity [15,16]. Therefore, the activity of new non-nucleoside antiviral drugs targeting viral polymerase against various viral infections is widely studied.

Previously, the antiviral activity of various non-nucleoside uracil derivatives was demonstrated. These compounds inhibited the replication of HCMV [17], VZV [18], HIV [19] and HCV [20]. Moreover, ramified derivatives of uracil presented antiviral activity against tick-borne encephalitis virus (TBEV) [21]. In the present study, we evaluated the antiviral activity of novel uracil derivatives against SARS-CoV-2 in vitro to determine their potential for further study as novel antiviral compounds against SARS-CoV-2 infection.

## 2. Results

### 2.1. Chemical Synthesis of Panel Compounds

The target acids were synthesized by treating the starting 1-(naphthalen-1-ylmethyl)uracil (**1**) [22], 1-(naphthalen-2-ylmethyl)uracil (**2**) [22], 1-(4-bromonaphthalen-1-ylmethyl)uracil (**3**) [22], 1-(anthrecen-9-ylmethyl)uracil (**4**) [23], 1-[3-(4-bromophenoxy)propyl]uracil (**5**) [24], 1-[5-(4-fluorophenoxy)pentyl]uracil (**6**), 1-[5-(2-bromophenoxy)pentyl]uracil (**7**) [25], 1-[5-(4-bromo-phenoxy)pentyl]uracil (**8**) [24], 1-[5-(3,5-dimethylphenoxy)pentyl]uracil (**9**) [26] or 1-[12-(4-bromophenoxy)dodecyl]uracil (**10**) [17] with an equimolar amount of 4-(ω-bromoalkoxy)benzoic acid methyl ester (**11–13**) in DMF solution in the presence of K_2_CO_3_. The resulting 1,3-disubstituted uracil without further purification was subjected to alkaline hydrolysis in an aqueous-alcoholic solution, which led to the formation of the corresponding benzoic acids (Figure 1).

In addition, we obtained a thymine derivative, the synthesis of which consisted of the treatment starting with 1-[5-(4-bromophenoxy)pentyl]thymine (**14**) [24] and an equimolar amount of 4-(5-bromopentyloxy)benzoic acid methyl ester (**13**) in DMF solution in the presence of K_2_CO_3_. The resulting 1,3-disubstituted thymine without further purification was subjected to alkaline hydrolysis in an aqueous-alcoholic solution, which produced the target benzoic acid **872** (Figure 2).

Next, we synthesized a uracil derivative linked to a benzoic acid residue by a methylene group. The synthesis of this compound was carried out by treatment starting with 1-[5-(4-bromo-phenoxy)pentyl]uracil (**8**) and an equimolar amount of 4-chloromethylbenzoic acid methyl ester (**15**) followed by alkaline hydrolysis in an aqueous-alcoholic medium. As a result, the target benzoic acid 875 was obtained at a 77% yield. (Figure 3).

We also obtained an analog of compound **876**, which did not contain a carboxyl group in the side chain. Its synthesis was carried out by the treatment of 1-(anthrecen-9-ylmethyl)uracil (**4**) with an equimolar amount of 1-bromo-4-[(6-bromohexyl)oxy]benzene (**16**) in DMF solution in the presence of K_2_CO_3_. In this case, the target compound **611** was formed, the yield of which was 66% (Figure 4).

### 2.2. Antiviral Activity

To study the antiviral effect of the compounds (Figure 1), Vero E6 cells were treated with the indicated concentrations and infected with the Delta (B.1.617.2) or Beta (B.1.351) variants of SARS-CoV-2. Antiviral activity was based on an analysis of the virus-induced cytopathic effect (CPE). The data obtained showed that the IC_50_ values ranged from 13.3 to 49.97 µM (Figure 1, Table 1). To determine the cytotoxicity of compounds, Vero E6 cells treated with test compounds were subjected to an MTT assay. The results demonstrate that most compounds do not significantly reduce cell viability and CC_50_ values were >50 μM. However, several compounds (**871**, **874** and **1007**) reduced cell proliferation without significant cell death.

We also confirmed the inhibitory effect of the compounds on the infection of the SARS-CoV-2 Omicron BA.1.1 using a plaque reduction assay (Figure 2). To assess their antiviral activity, half-maximal inhibitory doses (IC_50_) were determined. It is interesting to note that of the twelve compounds tested, only six showed a pronounced antiviral effect. The calculated IC_50_ values for these compounds ranged from 7.92 μM to 28.21 μM (Table 1).

To explore which steps of the SARS-CoV-2 replication cycle were interrupted by the test compounds, we performed a time-of-drug-addition assay for compounds **874** and **876**, which were selected as representatives with different substituents at positions N_1_ and N_3_. Cells were treated with 25 µm of these compounds at various stages of infection (entry, full time and post-entry) which was followed by qRT-PCR assays. The addition of compounds at the «entry» step did not inhibit the production of viral RNA (Figure 3). However, the addition of compounds at the «full time» step inhibited virus replication, resulting in a decrease in the viral load in the supernatant of infected Vero E6 cells. In addition, compound **876** reduced the viral load at the «post-entry» stage.

### 2.3. Mechanism of Action and Inhibition of SARS-CoV-2 RdRp

Molecular docking and the mean values of docking scores, as well as binding to SARS-CoV-2 RdRp were evaluated for all tested compounds. Due to the hydrophobic nature of uracil derivatives, the sites of interest are expected to be mostly hydrophobic too. One of such sites that may directly affect enzyme functions is the RNA cleft that features some non-polar cavities. This cleft is already being used as one of the docking targets by te hD3Pharma “D3Targets 2019-nCov” web server project [27]. The best score of 8.3 corresponds to **876** (Figure 4), so we docked it with an RNA cleft (Figure 5). Its closest anthracyl (**611**) and naphthyl (**1006** and **601**) analogs were predicted to have scores from -7.5 to -6.9, and the other compounds’ binding was worse.

The best binding poses of the leading compound **876** demonstrate the same structural feature, in that the naphthalene moiety was placed in mostly the hydrophobic pocket in the RNA cleft formed by VAL495, LEU498, LYS500, ALA512, VAL560, THR565, ARG569, and ILE572 (Figure 5a). The pocket corresponds well to the anthracene moiety size (Figure 5b). The remaining part of the **876** molecule demonstrates more diversity in poses but strongly tends to occupy the area of the cleft related to the template RNA strand of the enzyme apo form (Figure 5c). Polar contacts are also observable, e.g., the best scoring pose of **876** shown on Figure 5d reveals polar interactions with several residues, where the strongest ones appear between ligand carboxyl and THR687 (both side chain hydroxyl and backbone amino groups).

According to data showing that **876** can binding with SARS-CoV-2 RdRp, we measured its ability to inhibit the transcriptional activity of a purified SARS-CoV-2 RdRp composed of the NSP12 catalytic subunit and two additional proteins, NSP7 and NSP8. SARS-CoV-2 RdRp activity was inhibited by **876** at 100 µM and 50 µM, by 100% and 60%, respectively (Figure 6).

## 3. Discussion

The search for inhibitors of SARS-CoV-2 is an important challenge to combat the ongoing COVID-19 pandemic. The widespread use of vaccines to prevent COVID-19 has significantly reduced mortality from COVID-19 among the vaccinated, but the virus continues to circulate and some patients still require hospitalization. The ideal inhibitor of SARS-CoV-2 should be safe for widespread use and effective both early in the course of the disease and later on, reducing the severity of the disease and speeding up recovery. A desirable property would be the ability to reduce the transmission of the virus in a population. In addition, it must have a high threshold of resistance and activity against all circulating strains.

The drugs recommended for treatment with a specific antiviral mechanism of action, including remdesivir, molnupiravir, and nirmatrelvir, were not developed specifically as inhibitors of SARS-CoV-2 and were proposed during the pandemic using the repurposing mechanism with minimal modifications. They do not sufficiently satisfy the above criteria and have a number of significant drawbacks. Remdesivir, which has a terminating mechanism of action, is an injectable drug used in the clinic to reduce the severity of the disease [28]. The mechanism of action of molnupiravir lies in the mutagenesis of the viral genome, which causes the risks of accelerating the evolution of the virus with its widespread use, as well as its toxicity [29]. Nirmatrelvir, being the most extensively redesigned to inhibit SARS-CoV-2, showed the highest activity [30]. However, the peptide nature of the drug does not allow us to rely on its wide availability for the treatment of patients with COVID-19 in the near future.

The group of non-nucleoside inhibitors studied by us, which are N_1_,N_3_-disubstituted derivatives of uracil, is distinguished by the simplicity of chemical synthesis. The study of the mechanism of action, taking into account the data of molecular docking and the assessment of polymerase activity in vitro, indicates the ability to inhibit the RNA groove of the RdRp (Nsp12). Taking into account the high conservatism of Nsp12, the manifestation of the inhibitory activity of the compounds against various genetic variants of the virus is not surprising.

The disadvantage of the studied group of N_1_,N_3_-disubstituted uracil derivatives is the low solubility. Our current efficacy studies using an animal model indicate the need to create a stabilized finished dosage form, leading to an increase in the solubility and bioavailability of the target compounds. Thus, the main tasks in working with this group of compounds are to obtain analogs with increased solubility and the stabilization of molecules in the process of obtaining finished dosage forms for oral administration.

## 4. Materials and Methods

### 4.1. General

All reagents were obtained from Sigma and Acros Organics. Anhydrous 1,2-dichloroethane and EtOAc were obtained by distillation over P_2_O_5_. TLC was performed on Merck TLC Silica gel 60 F_254_ plates eluted with the specified solvents and samples were made visual with a UV lamp, VL-6. LC (Vilber Lourmat, Eberald Zell, Germany). Acros Organics (Acros Organics BV, Geel, Belgium) silica gel (Kieselgur 60–200 µm, 60 A) was used for column chromatography. Yields refer to spectroscopically (^1^H and ^13^C NMR) homogeneous material s. Melting points were determined in glass capillaries on a Mel-Temp 3.0 (Laboratory Devices Inc., Auburn, CA, USA).

### 4.2. General Procedure for the Synthesis of Acids ***601***, ***870***, ***871***, ***872***, ***873*** and ***874***

A mixture 1.416 mmol of 1-substituted uracil (**1**–**10**) and 0.29 g (2.098 mmol) K_2_CO_3_ in a solution of 10 mL of DMF was stirred at 80 °C for 1 h, and 1.40 mmol of 4-(ω-bromoalkoxy) benzoic acid methyl ester (**11**–**13**) was added and stirred at the same temperature for 24 h. Then, the reaction mass was evaporated in a vacuum, water (100 mL) was added to the residue, it was extracted with 1,2-dichloroethane (4 × 25 mL), and the extract was evaporated under reduced pressure. The residue was purified by flash chromatography followed by evaporation of the eluent under reduced pressure. The residue was dissolved in a mixture of ethanol (50 mL) and water (30 mL), 0.3 g (7.500 mmol) of NaOH was added, and the resulting mixture was stirred at room temperature for two days. Ethanol was evaporated under reduced pressure, the residue was diluted with water (200 mL) and then it was acidified with hydrochloric acid to pH 2. The precipitate was filtered off, dried in air, and crystallized from an ethyl acetate–hexane mixture (2:1).

NMR spectra were obtained using Bruker Avance 400 (400 MHz for ^1^H and 100 MHz for ^13^C) spectrometer in DMSO-d6 or CDCl3 with tetramethylsilane as an internal standard.

4-[4-[2,6-Dioxo-3-(naphthyl-1-methyl)-3,6-dihydropyrimidin-1(^2^*H*)-yl] butoxy]benzoic acid (**601**). Yield 76%, mp 183–184.5 °C, R_f_ 0.52 (i-PrOH-ethyl acetate-NH4OH, 9:6:5); ^1^H NMR (400 Hz, CDCl_3_), *δ*, ppm: 1.67 (^2^H, quin, J = 8.2 Hz, CH_2_), 1.88 (^2^H, quin, J = 7.9 Hz, CH_2_), 4.01–4.08 (^4^H, m, CH_2_ × 2), 5.32 (^2^H, s, CH_2_), 5.64 (^1^H, d, J = 8.1 Hz, uracil H-5), 6.91 (^1^H, d, J = 8.9 Hz, H-3′, H-5′), 6.99 (^1^H, d, J = 7.9 Hz, uracil H-6), 7.36–7.48 (^2^H, m, aromatic H), 7.50–7.57 (^2^H, m, aromatic H), 7.87–7.95 (^3^H, m, aromatic H), 8.03 (^2^H, d, J = 8.9 Hz, H-2′, H-6′). ^13^C NMR (100 MHz, CDCl_3_), *δ*, ppm: 23.5, 27.4, 41.4, 49.3, 68.0, 102.4, 112.7, 116.4, 123.4, 124.7, 127.7, 128.0, 128.2, 128.5, 129.5, 130.7, 132.3, 132.5, 140.5, 151.8, 158.3, 162.7.

4-[5-[3-[5-(4-Fluorophenoxy)pentyl]-2,6-dioxo-3,6-dihydropyrimidin-1(^2^*H*)-yl]pentyloxy]benzoic acid (**870**). Yield 88%, mp 133–135 °C, R_f_ 0.54 (i-PrOH-ethyl acetate-NH_4_OH, 9:6:5); ^1^H NMR (400 Hz, DMSO-d_6_), *δ*, ppm: 1.36–1.39 (^4^H, m, CH_2_ × 2), 1.55–1.75 (^8^H, m, CH_2_ × 4), 3.69–4.00 (^8^H, m, CH_2_ × 4), 5.66 (^1^H, d, J = 7.8 Hz, uracil H-5), 6.87 (^2^H, d, J = 9 Hz, H-3′, H-5′), 6.90 (^2^H, d, J = 8.9 Hz, H-3″, H-5″), 6.97 (^2^H, d, J = 7.9 Hz, H-2″, H-6″), 7.68 (^1^H, d, J = 7.8 Hz, uracil H-6), 7.86 (^2^H, d, J = 8.9 Hz, H-2′, H-6′). ^13^C NMR (100 MHz, DMSO-d_6_), *δ*, ppm: 22.4, 22.8, 26.8, 28.1, 28.2, 28.3, 48.5, 67.6, 67.7, 100.1, 114.2, 115.5, 115.6, 115.7, 115.9, 122.8, 131.4, 144.1, 151.0, 155.0, 155.2, 157.6, 162.3, 162.4, 167.1.

4-[5-[3-[5-(2-Bromophenoxy)pentyl]-2,6-dioxo-3,6-dihydropyrimidin-1(^2^*H*)-yl]pentyloxy]benzoic acid (**871**). Yield 84%, mp 91–92 °C, R_f_ 0.55 (i-PrOH-ethyl acetate-NH_4_OH, 9:6:5); ^1^H NMR (400 Hz, DMSO-d_6_), *δ*, ppm: 1.36–1.45 (^4^H, m, CH_2_ × 2), 1.61–1.77 (^8^H, m, CH_2_ × 4), 3.70–4.01 (^8^H, m, CH_2_ × 4), 5.65 (^1^H, d, J = 7.8 Hz, uracil H-5), 6.84 (^1^H, dt, J = 6.8 and 1.2 Hz, H-4′), 6.96 (^1^H, d, J = 8.8 Hz, H-3″, H-5″), 7.04 (^1^H, dd, J = 8.2 and 1.3 Hz, H-6′), 7.28 (^1^H, dt, J = 1.6 and 7.0 Hz, H-5′), 7.52 (^1^H, dd, J = 7.9 and 1.6 Hz, H-3′), 7.67 (^1^H, d, J = 7.8 Hz, uracil H-6), 7.86 (^2^H, d, J = 8.8 Hz, H-2″, H-6″). ^13^C NMR (100 MHz, DMSO-d_6_), *δ*, ppm: 22.4, 22.8, 26.8, 28.0, 28.1, 28.2, 48.5, 67.6, 68.2, 100.1, 111.1, 113.7, 114.2, 121.9, 122.8, 129.0, 131.4, 132.9, 144.1, 151.0, 154.8, 162.3, 162.4, 167.1.

4-[5-[3-[5-(2-Bromophenoxy)pentyl]-2,6-dioxo-5-methyl-3,6-dihydropyrimidin-1(^2^*H*)-yl]pentyloxy]benzoic acid (**872**). Yield 81%, mp 132–134 °C, R_f_ 0.53 (i-PrOH-ethyl acetate-NH_4_OH, 9:6:5); ^1^H NMR (400 Hz, DMSO-d_6_), *δ*, ppm: 1.38–1.46 (^4^H, m, CH_2_ × 2), 1.60–1.75 (^8^H, m, CH_2_ × 4), 3.62–4.08 (^8^H, m, CH_2_ × 4), 6.89 (^2^H, d, J = 9 Hz, H-3′, H-5′), 6.91 (^2^H, d, J = 8.9 Hz, H-3″, H-5″), 6.99 (^2^H, d, J = 7.9 Hz, H-2″, H-6″), 7.42 (^2^H, d, J = 9.1 Hz, H-2′, H-6′), 7.53 (^1^H, s, H-6). ^13^C NMR (100 MHz, DMSO-d_6_), *δ*, ppm: 12.8, 22.7, 22.9, 26.9, 28.1, 28.2, 28.4, 48.5, 67.6, 67.7, 100.1, 114.2, 115.5, 115.6, 115.7, 115.9, 122.8, 131.4, 144.1, 151.0, 155.0, 155.2, 157.6, 162.3, 162.4, 167.1.

4-[4-[3-[3-(4-Bromophenoxy)propyl]-2,6-dioxo-3,6-dihydropyrimidin-1(^2^*H*)-yl]butoxy]benzoic acid (**873**). Yield 79%, mp 122–124 °C, R_f_ 0.51 (i-PrOH-ethyl acetate-NH_4_OH, 9:6:5); ^1^H NMR (400 Hz, DMSO-d_6_), *δ*, ppm: 1.58 (^2^H, quin, J = 7.3 Hz, CH_2_), 1.66 (2H, quin, J = 7.5 Hz, CH_2_), 2.01 (^2^H, quin, J = 6.2 Hz, CH_2_), 3.69 (^2^H, t, J = 7.1 Hz, NCH_2_), 3.89 (^2^H, t, J = 6.4 Hz, OCH_2_), 3.99 (^2^H, t, J = 6.7 Hz, N(3)CH_2_), 4.09 (^2^H, t, J = 5.9 Hz, OCH_2_), 5.53 (^1^H, d, J = 7.8 Hz, uracil H-5), 6.72 (^1^H, d, J = 8.8 Hz, H-3″, H-5″), 6.87 (^2^H, d, J = 9.0 Hz, H-3′, H-5′), 7.04 (^2^H, d, J = 8.8 Hz, H-2″, H-6″), 7.43 (^2^H, d, J = 8.9 Hz, H-2′, H-6′), 7.62 (^1^H, d, J = 7.8 Hz, uracil H-6). ^13^C NMR (100 MHz, DMSO-d_6_), *δ*, ppm: 22.9, 26.9, 28.4, 31.1, 44.9, 49.9, 68.6, 103.3, 115.4, 120.1, 122.8, 123.5, 125.5, 131.4, 135.5, 148.2, 154.3, 161.0, 165.4, 172.5.

4-[5-[3-[5-(3,5-Dimethylphenoxy)pentyl]-2,6-dioxo-3,6-dihydropyrimidin-1(^2^*H*)-yl]pentyloxy]benzoic acid (**874**). Yield 81%, mp 103.5–105 °C, R_f_ 0.57 (i-PrOH-ethyl acetate-NH_4_OH, 9:6:5); ^1^H NMR (400 Hz, DMSO-d_6_), *δ*, ppm: 1.34–1.38 (^4^H, m, CH_2_ × 2), 1.52–1.73 (^8^H, m, CH_2_ × 4), 2.17 (^6^H, s, CH_3_ × 2), 3.68–3.98 (^8^H, m, CH_2_ × 4), 5.64 (^1^H, d, J = 7.9 Hz, uracil H-5), 6.47 (^2^H, s, H-2′, H-6′), 6.49 (^1^H, s, H-4′), 6.95 (^1^H, d, J = 8.9 Hz, H-3″, H-5″), 7.64 (^1^H, d, J = 7.8 Hz, uracil H-6), 7.85 (^2^H, d, J = 8.8 Hz, H-2″, H-6″). ^13^C NMR (100 MHz, DMSO-d_6_), *δ*, ppm: 21.1, 22.5, 26.8, 28.1, 28.2, 28.4, 47.5, 67.6, 67.7, 100.8, 114.2, 115.5, 115.6, 115.7, 115.9, 122.8, 131.4, 144.1, 151.0, 155.0, 155.2, 157.6, 162.3, 162.4, 167.1.

4-[[5-[3-(Anthracen-9-ylmethyl)-2,6-dioxo-3,6-dihydropyrimidin-1(^2^*H*)-yl]pentyloxy]benzoic acid (**876**). Yield 76%, mp 173–176 °C, R_f_ 0.52 (i-PrOH-ethyl acetate-NH_4_OH, 9:6:5); ^1^H NMR (400 Hz, CDCl_3_), *δ*, ppm: 1.31 (^2^H, quin, J = 6.8 Hz, CH_2_), 1.59 (^2^H, quin, J = 7.6 Hz, CH_2_), 1.67 (^2^H, quin, J = 7.6 Hz, CH_2_), 3.54 (^2^H, t, J = 7.1 Hz, NCH_2_), 3.77 (^2^H, t, J = 6.1 Hz, OCH_2_), 5.68 (^1^H, d, J = 7.8 Hz, uracil H-5), 6.13 (^2^H, s, CH_2_), 6.72 (^1^H, d, J = 8.8 Hz, H-3′, H-5′), 6.92 (^1^H, t, J = 7.6 Hz, uracil H-6), 7.00 (^2^H, d, J = 8.8 Hz, H-2′, H-6′), 7.45 (^2^H, t, J = 6.8 Hz, H-3″, H-6″), 7.54 (^2^H, t, J = 6.8 Hz, H-2″, H-7″), 7.98 (^2^H, d, J = 8.3 Hz, H-1″, H-8″), 8.41 (^1^H, s, H-10″), 8.53 (^2^H, d, J = 9.1 Hz, H-4″, H-5″). ^13^C NMR (100 MHz, DMSO-d_6_), *δ*, ppm: 22.4, 23.8, 26.1, 42.5, 45.1, 67.5, 101.0, 114.3, 122.9, 123.5, 125.5, 127.6, 129.4, 130.9, 131.1, 131.4, 141.0, 151.5, 162.0, 162.3, 167.1.

4-[4-[2,6-Dioxo-3-(naphthyl-2-methyl)-3,6-dihydropyrimidin-1(^2^*H*)-yl]butoxy]benzoic acid (**1005**). Yield 72%, mp 143.5–145 °C, R_f_ 0.52 (i-PrOH-ethyl acetate-NH_4_OH, 9:6:5); ^1^H NMR (400 Hz, DMSO-d_6_), *δ*, ppm: 1.66–1.70 (^4^H, m, CH_2_ × 2), 3.86 (^2^H, t, J = 7.3 Hz, NCH_2_), 4.00 (^2^H, t, J = 6.4 Hz, OCH_2_), 5.09 (^2^H, s, ArCH_2_), 5.76 (^1^H, d, J = 7.9 Hz, uracil H-5), 6.95 (^2^H, d, J = 9.0 Hz, H-3′, H-5′), 7.43–7.50 (^3^H, m, aromatic H), 7.79 (^1^H, s, H-1″), 7.85–7.89 (^6^H, m, J = 7.9 Hz, uracil H-6, aromatic H). ^13^C NMR (100 MHz, DMSO-d_6_), *δ*, ppm: 23.9, 26.1, 51.7, 67.4, 100.8, 114.2, 122.9, 125.5, 126.2, 126.5, 127.6, 127.8, 128.4, 131.4, 132.4, 132.8, 134.3, 144.2, 151.3, 162.2, 162.5, 167.1.

4-[4-[3-(4-Bromonaphthyl-1-methyl)-2,6-dioxo-3,6-dihydropyrimidin-1(2*H*)-yl]butoxy]benzoic acid (**1006**). Yield 69%, mp 195.5–197.5 °C, R_f_ 0.52 (i-PrOH-ethyl acetate-NH_4_OH, 9:6:5); 1H NMR (400 Hz, CDCl_3_), *δ*, ppm: 1.73 (^2^H, quin, J = 8.0 Hz, CH_2_), 1.88 (^2^H, quin, J = 7.9 Hz, CH_2_), 3.92 (^2^H, t, J = 6.4 Hz, NCH_2_), 4.03 (^2^H, t, J = 7.5 Hz, OCH_2_), 5.34 (^2^H, s, ArCH_2_), 5.63 (^1^H, d, J = 8.0 Hz, uracil H-5), 6.75 (^1^H, d, J = 9.1 Hz, H-3′, H-5′), 6.97 (^1^H, d, J = 7.9 Hz, aromatic H), 7.19 (^1^H, d, J = 7.6 Hz, aromatic H), 7.34 (^2^H, d, J = 9.0 Hz, H-2′, H-6′), 7.57–7.65 (^2^H, m, aromatic H), 7.76 (^1^H, d, J = 7.7 Hz, uracil H-6), 7.94 (^1^H, dd, J = 7.7 and 1.4 Hz, aromatic H), 8.32 (^1^H, dd, J = 7.8 and 1.4 Hz, aromatic H). ^13^C NMR (100 MHz, DMSO-d_6_), *δ*, ppm: 23.5, 27.4, 28.9, 49.3, 68.0, 102.4, 112.7, 116.4, 123.4, 124.7, 127.7, 128.0, 128.2, 128.5, 129.5, 130.7, 132.3, 132.5, 140.5, 151.8, 158.3, 162.7.

4-[3-[3-[12-(4-Bromophenoxy)dodecyl]-2,6-dioxo-3,6-dihydropyrimidin-1(^2^*H*)-yl]propoxy]benzoic acid (**1007**). Yield 54%, mp 110–112 °C, R_f_ 0.56 (i-PrOH-ethyl acetate-NH_4_OH, 9:6:5); ^1^H NMR (400 Hz, CDCl_3_), *δ*, ppm: 1.23–1.31 (^16^H, m, CH_2_ × 8), 1.34–1.39 (^6^H, m, CH_2_ × 3), 3.74–3.77 (^4^H, m, CH_2_ × 2), 3.91–3.94 (^4^H, m, CH_2_ × 2), 5.71 (^1^H, d, J = 7.9 Hz, uracil H-5), 6.88 (^1^H, d, J = 9.1 Hz, H-3′, H-5′), 7.41 (^2^H, d, J = 9.1 Hz, H-2′, H-6′), 7.64 (^1^H, d, J = 7.8 Hz, uracil H-6). ^13^C NMR (100 MHz, CDCl_3_), *δ*, ppm: 22.6, 25.4, 26.2, 26.8, 27.0, 28.5, 28.6, 28.7, 28.9, 49.6, 67.7, 100.1, 111.6, 116.7, 128.0, 128.2, 128.4, 129.5, 132.0, 132.3, 132.5, 140.5, 144.4, 151.0, 157.9, 162.3, 169.3.

4-[[5-[3-[5-(4-Bromophenoxy)pentyl]-2,6-dioxo-3,6-dihydropyrimidin-1(^2^*H*)-yl]methyl]benzoic acid (**875**). A mixture of 0.5 g (1.416 mmol) of 1-[5-(4-bromophenoxy)pentyl]uracil (**8**) and 0.3 g (2.171 mmol) of K_2_CO_3_ in DMF solution (10 mL) was stirred at 80 °C for 1 h, cooled to room temperature, and 0.32 g (1.397 mmol) of 4-chloromethylbenzoic acid methyl ester (**15**) was added and stirred at room temperature for 24 h. Then, the reaction mass was evaporated in a vacuum, the residue was treated with 100 mL of water, extracted with 1,2-dichloroethane (4 × 25 mL) and the extract was evaporated under reduced pressure. The residue was purified by performing flash chromatography, followed by evaporation of the eluent under reduced pressure. The residue was dissolved in a mixture of ethanol (50 mL) and water (30 mL), 0.3 g (7.5 mmol) of NaOH was added, and the resulting mixture was stirred at room temperature for two days. Ethanol was evaporated under reduced pressure, the residue was diluted with water (200 mL) and acidified with hydrochloric acid to pH 2. The precipitate that formed was filtered, dried in air, and the product was crystallized from an ethyl acetate–hexane mixture. (3:1). Yield 77%, mp 186–187 °C, R_f_ 0.52 (i-PrOH-ethyl acetate-NH_4_OH, 9:6:5); ^1^H NMR (400 Hz, DMSO-d_6_), *δ*, ppm: 1.36 (^2^H, quin, J = 8.1 Hz, CH_2_), 1.61 (^2^H, quin, J = 7.3 Hz, CH_2_), 1.70 (^2^H, quin, J = 7.6 Hz, CH_2_), 3.65 (^2^H, t, J = 7.2 Hz, NCH_2_), 3.92 (^2^H, t, J = 6.5 Hz, OCH_2_), 5.04 (^2^H, s, ArCH_2_), 5.53 (^1^H, d, J = 7.8 Hz, uracil H-5), 6.76 (^1^H, d, J = 9.1 Hz, H-3′, H-5′), 7.34 (^1^H, d, J = 8.3 Hz, H-3″, H-5″), 7.41 (^2^H, d, J = 8.8 Hz, H-2′, H-6′), 7.64 (^1^H, d, J = 7.8 Hz, uracil H-6), 7.88 (^2^H, d, J = 8.3 Hz, H-2″, H-6″). ^13^C NMR (100 MHz, DMSO-d_6_), *δ*, ppm: 22.4, 28.2, 47.3, 50.2, 67.5, 100.8, 111.8, 116.7, 132.1, 132.5, 134.4, 145.7, 151.0, 157.9, 163.8, 169.6.

1-(Anthracen-9-ylmethyl)-3-[6-(4-bromophenoxy)hexyl]uracil (**611**). A mixture of 0.5 g (1.643 mmol) 1-(anthracen-9-ylmethyl)uracil (**4**) and 0.29 g (2.098 mmol) K_2_CO_3_ in a DMF (10 mL) solution was stirred at 80 °C for 1 h, 0.56 g (1.666 mmol) of 1-bromo-(6-bromohexyloxy)benzene methyl ester (**16**) and stirred at the same temperature for 24 h. Then, the reaction mass was evaporated in a vacuum, the residue was extracted with 1,2-dichloroethane (4 × 25 mL) and the extract was evaporated under reduced pressure. The residue was purified by performing flash chromatography, followed by evaporation of the eluent under reduced pressure. The residue was crystallized from ethyl acetate–hexane (3:1). Yield 66%, mp 109.5–111 °C, R_f_ 0.68 (1,2-dichloroethane-ethyl acetate, 1:1); ^1^H NMR (400 Hz, CDCl_3_), *δ*, ppm: 1.31 (^2^H, quin, J = 6.8 Hz, CH_2_), 1.59 (^2^H, quin, J = 7.6 Hz, CH_2_), 1.67 (^2^H, quin, J = 7.6 Hz, CH_2_), 3.54 (^2^H, t, J = 7.1 Hz, NCH_2_), 3.77 (^2^H, t, J = 6.1 Hz, OCH_2_), 5.68 (^1^H, d, J = 7.8 Hz, uracil H-5), 6.13 (^2^H, s, CH_2_), 6.72 (^1^H, d, J = 8.8 Hz, H-3′, H-5′), 6.93 (^1^H, t, J = 7.6 Hz, uracil H-6), 7.37 (^2^H, d, J = 8.8 Hz, H-2′, H-6′), 7.45 (^2^H, t, J = 6.8 Hz, H-3″, H-6″), 7.54 (^2^H, t, J = 6.8 Hz, H-2″, H-7″), 7.98 (^2^H, d, J = 8.3 Hz, H-1″, H-8″), 8.41 (^1^H, s, H-10″), 8.53 (^2^H, d, J = 9.1 Hz, H-4″, H-5″). ^13^C NMR (100 MHz, CDCl_3_), *δ*, ppm: 22.7, 27.4, 28.47, 28.51, 38.4, 49.2, 67.4, 101.3, 112.6, 116.1, 124.6, 124.7, 125.8, 127.9, 128.2, 129.0, 131.0, 131.2, 132.1, 142.1, 151.3, 157.9, 163.5.

### 4.3. Cells and Viruses

African green monkey kidney Vero E6 cells (ATCC^®^-1586) were propagated in DMEM (Gibco, Gaithersburg, MD, USA) supplemented with 1% (*v*/*v*) penicillin/streptomycin solution (Gibco, USA) and heat-inactivated 10% (*v*/*v*) fetal bovine serum (FBS) (HyClone, Logan, UT, USA). In this study, we used the following SARS-CoV-2 strains: the Delta variant B.1.617.2 (hCoV-19/Russia/MOW-Moscow_PMVL-49/2021; EPI_ISL_4572812), the Beta variant B.1.351 (hCoV-19/Netherlands/NoordHolland_10159/2021; Ref-SKU: 014V-04058), and the Omicron variant BA.1 (hCoV-19/Russia/MOW-Moscow_PMVL-O16/2021; EPI_ISL_7263933). The viruses were isolated from oro/nasopharyngeal swabs and propagated in Vero E6. All experiments using infectious SARS-CoV-2 were performed in a biosafety level 3 (BSL3) laboratory.

### 4.4. Cytopathic Effect Inhibition Antiviral Assay

Vero E6 cells (2 × 10^4^ cells/well) were seeded into 96-wells plates and treated with different concentrations of test compounds (1:4 serial dilutions, from 100 to 0.097 µM). Each compound concentration was evaluated for both antiviral efficacy and cytotoxicity. Then, the cells were infected with SARS-CoV-2 at 100TCID_50_. Cell cultures were incubated at 37 °C in 5% CO_2_ for 72 h prior to assessment of the virus-induced cytopathic effect (CPE). CPE and cytotoxicity of the compound were determined using an MTT assay as described recently [31].

### 4.5. Plaque Reduction Assay

A plaque reduction assay was performed to plot the 50% inhibitory concentration (IC_50_) of individual compounds against the Omicron variant of SARS-CoV-2 as described in [32] with some modifications. Vero E6 cells were seeded at 3 × 10^4^ cells per well in 96-well plates the day before the assay. After 18 h of incubation, 50 p.f.u. SARS-CoV-2 in the presence or absence of compounds was added to the cell monolayer, and the plates were incubated for an additional 1 h at 37 °C in 5% CO_2_ before removing unbound viral particles by removing the medium and washing with PBS. Monolayers were then overlaid with DMEM containing 0.7% carboxymethylcellulose (CMC, Sigma, St. Louis, MO, USA) with appropriate concentrations of the individual compounds, and then incubated for an additional 72 h. Then, the wells were fixed with 10% formaldehyde and CMC was removed. Monolayers were stained with 0.5% crystal violet (Sigma, St. Louis, MO, USA) and plaques were counted. The percent of inhibition of plaque formation relative to control wells (without compounds) was determined for each test compound concentration.

### 4.6. Time-of-Addition Experiments

The assays were performed as described previously [31]. Briefly, Vero E6 cells were seeded into 96-well plates and treated with **874** or **876** (25 µM) at different stages of virus infection (full-time, entry and post-entry). The cells were infected with SARS-CoV-2 (MOI  =  0.01) and then incubated for 1 h. The viral inoculum was then removed, and the cells were washed twice with PBS. At 18 h post-infection, the cell culture supernatant of each time point experiment was collected for viral yield measurements using qRT–PCR, as described in [31,33].

### 4.7. Molecular Docking

Molecular docking studies were performed on the set of compounds for the SARS-CoV-2 RdRp main protein (NSP12) using Smina software [34] and an AutoDock Vina fork [35]. Several structures of SARS-CoV-2 RdRp are currently available on RCSB.org, including different combinations of cofactors (replicative complex member proteins, RNA and ligands). In our study, we used the NSP12 structure from PDB ID 7EIZ (CryoEM structure of the replicative complex including RNA duplex fragment) [36]. We removed the RNA duplex from the cleft before docking. A ~9700 A3 grid box was used, including the cleft itself. Chain A (NSP12) from 7EIZ was prepared using the pdb2pqr 3.0 tool [37,38] with default parameters (pH 7.0, AMBER force field). Then, it was converted to a pdbqt format using the prepare_receptor script from ADFRsuite version 1.0 [39]. Ligands were converted to the pdbqt format using the prepare_ligand script included in the same suite. The docking procedure was repeated 4 times with a random initial seed to gather statistics. Poses with best score values (kcal/mol) were taken from each run. Structures were visualized using PyMol [40].

### 4.8. Inhibition Activity of SARS-CoV-2 RNA-Dependent RNA Polymerase (RdRp)

The activity of SARS-CoV-2 RdRp was assessed using the SARS-CoV-2 RNA-dependent RNA polymerase kit plus (Profoldin #S2RPA100KE), according to the manufacture instructions. In brief, reactions were carried out at 35 °C for 2 h in the presence or absence (control) of the test compound. Then, fluorescent dye was added, and fluorescence was determined using a Qubit fluorimeter (Thermo Fisher, MA, USA). The inhibition of RdRp activity was calculated from the values obtained for control samples.

## 5. Conclusions

Our study demonstrated in vitro efficacy novel uracil derivatives against SARS-CoV-2 VOCs, including «Omicron». We also found that compound **876** was able to directly inhibit the activity of RdRp. The presented docking results reflect one possible mechanism of RdRp inhibition by the **876** compound. Other compounds also demonstrated the possibility to dock into this pocket but with worse score values. Anthracyl analogs demonstrated lower RMSD between poses and a more advanced fit, suggesting that such a class of compounds may be promising for the design of further SARS-CoV-2 inhibitors. We expect that our findings will serve as a starting point for further testing of the selected candidates in more complex and biologically meaningful preclinical models of SARS-CoV-2 infection as potential antivirals.

## 6. Patents

Uracil Derivatives with Antiviral Activity Against SARS-CoV-2 RU 2 769 828 (28.12.2021).

## Data Availability

Not applicable.

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
