# Peer review of "Antiviral Activity of N1,N3-Disubstituted Uracil Derivatives against SARS-CoV-2 Variants of Concern"

_ijms, 2022, doi:10.3390/ijms231710171_

Round 1

Reviewer 1 Report

The authors evaluated the activity of uracil derivatives against SARS-CoV-2, obtaining non-nucleoside analogs with a putative antiviral effect. Regards this work, my recommendations are shown as follows:

  1. Authors need to remark on some published works about some pharmacophoric models and de novo-designed molecules against the SARS-CoV-2, for example, https://doi.org/10.1039/D1CP04159B and https://doi.org/10.4155/fmc-2020-0262.
  2. The docking assay lacks discussion about the non-covalent interaction between the ligands and the selected target. The authors only mention the interacting residues but do not specify the kind of interactions.
  3. As well as, the authors need to add a comparison of computational results with some reported in the literature.
  4. Conclusions and abstract do not specify the interaction energy for the best-studied ligand and the selected target.

Author Response

  1. Reviewer: Authors need to remark on some published works about some pharmacophoric models and de novo-designed molecules against the SARS-CoV-2, for example, https://doi.org/10.1039/D1CP04159B and https://doi.org/10.4155/fmc-2020-0262.

Response: Thank you for providing links for these works. We should note that our molecular docking study has mostly illustrative purpose and actually is some kind of speculation about one possible mechanism of action. We are not positioning it as a conclusion or hypothesis about even most realistic way how these compounds work on replicating virus. Therefore we are not performing exhaustive research here.

  1. Reviewer: The docking assay lacks discussion about the non-covalent interaction between the ligands and the selected target. The authors only mention the interacting residues but do not specify the kind of interactions.

Response: We have provided more detailed information about contacts between RdRp and ligand (876) (386-389, Fig. 5).

  1. Reviewer: As well as, the authors need to add a comparison of computational results with some reported in the literature.

Response: Available literature lacks data about ligands directly binding to RNA cleft pockets of SARS-CoV-2. Nevertheless, we know that D3Pharma project (https://www.d3pharma.com/D3Targets-2019-nCoV/) use such docking target in their calculations, so we referred to them (372-373).

  1. Reviewer: Conclusions and abstract do not specify the interaction energy for the best-studied ligand and the selected target.

Response: The docking score represents calculated binding energy itself (kcal/mol, please see "Material and Methods", "Molecular docking").

Reviewer 2 Report

I would like to congratulate with the authors for the manuscript they have put together. In this study, they evaluated the therapeutic potential of uracile-derived compounds in antagonizing the SARS-CoV-2 VOCs like Omicron. They found that compound 876 was able to inhibit the RNA groove of the Nsp12 but shows low solubility, which raises the need to produce analogs with higher solubility and stability.

The manuscript is well written in all its parts, the results are clearly presented and discussed. For the reasons above, I endorse this manuscript for publication in IJMS. 

Our study demonstrated in vitro efficacy novel uracil derivatives against SARS- 435 CoV-2 VOCs, including «Omicron». We also found that compound 876 was able to di- 436 rectly inhibit activity of RdRp

Author Response

We are grateful to the reviewer for a good review of our article.